# Diffuse Sound Absorptive Properties of Parallel-Arranged Perforated Plates with Extended Tubes and Porous Materials

**DOI:** 10.3390/ma13051091

**Published:** 2020-03-01

**Authors:** Dengke Li, Daoqing Chang, Bilong Liu

**Affiliations:** 1The State Key Laboratory of Heavy Duty AC Drive Electric Locomotive Systems Integration, Zhuzhou 412001, China; 2Key Laboratory of Noise and Vibration Research, Institute of Acoustics, Chinese Academy of Sciences, Beijing 100190, China; 3School of Mechanical and Automotive Engineering, Qingdao University of Technology, No. 777 Jialingjiang Road, Qingdao 266520, China; Liubilong@qut.edu.cn

**Keywords:** sound absorption, perforated plates with extended tubes, porous materials, periodic absorber

## Abstract

The diffuse sound absorption was investigated theoretically and experimentally for a periodically arranged sound absorber composed of perforated plates with extended tubes (PPETs) and porous materials. The calculation formulae related to the boundary condition are derived for the periodic absorbers, and then the equations are solved numerically. The influences of the incidence and azimuthal angle, and the period of absorber arrangement are investigated on the sound absorption. The sound-absorption coefficients are tested in a standard reverberation room for a periodic absorber composed of units of three parallel-arranged PPETs and porous material. The measured 1/3-octave band sound-absorption coefficients agree well with the theoretical prediction. Both theoretical and measured results suggest that the periodic PPET absorbers have good sound-absorption performance in the low- to mid-frequency range in diffuse field.

## 1. Introduction

Micro-perforated panels (MPP) are widely used in the engineering noise control of vehicles, buildings, and ventilation facilities, because of their good sound-absorption performance and many advantages, such as environmental friendliness, durability, and hygienic properties. In 1975, Maa [1] proposed the idea that panels with sub-millimeter perforations could provide sufficient acoustic resistance to achieve a high absorption coefficient. At the same time, he founded the theory of MPP and presented some engineering designs. Although the MPP sound absorber has a larger frequency band of absorption compared to conventional perforated panels, it does not satisfy some engineering requirements, especially in the low frequency range. Maa [2] further introduced a double-layer MPP to increase the bandwidth of sound absorption. 

In 1992, micro-perforated acrylic–glass plates were successfully used to attenuate the reverberation sound in German Bundestag Hall [3]. Later many researchers tried to improve the sound-absorptive properties of MPP, using various methods. When the impedance of the pores is comparable with the impedance of the panel or membrane on which the pores go through, their coupling effect should be considered. Kang et al. [4] presented a theoretical method to predict the sound-absorption coefficient of such structure. With appropriate parameters, the absorption coefficient can exceed 0.4 over 3–4 octaves. Lee et al. [5] theoretically studied the sound absorption of finite flexible micro-perforated panel, results show that the additional sound-absorption peak of the panel vibration effect can widen the absorption bandwidth of MPP when the resonant frequency of the flexible panel is higher than that of Helmholtz-resonance frequency of the MPP. Sakagami et al. [6] also used an electro-acoustical equivalent circuit model to study the relationship between the panel-type absorbers and MPP absorbers. Further research by Toyoda et al. [7] revealed that only the panel-type absorption caused by the eigen-mode vibration can occur independently from the Helmholtz-resonance absorption of MPP, and that the panel-type absorption caused by a mass-spring resonance cannot be utilized to widen the absorption bandwidth. Sakagami et al. [8,9] proposed a double-leaf MPP spatial sound absorber (DLMPP) which is composed of two MPP leaves, without the backing structure. A triple layer absorber composed of permeable membrane (PM) [10] and the double-leaf MPP (DLMPP) was proposed to improve the absorption performance, and the peak absorption at resonance was found to become significantly higher. The effect of a honeycomb in the air cavity was also discussed on the proposed absorber: The resonance peak is shifted to lower frequencies, and the level of absorption becomes higher at mid and high frequencies, owing to the effect of the honeycomb. Through an experiment and numerical simulation, Liu et al. [11] demonstrated that partitioning the adjoining cavity enhances the attenuation of the acoustic modes propagating transverse to the MPP. This effect was particularly noticeable at low frequencies where the acoustic response is resonant in nature. Wang et al. [12] used parallel-arranged MPPs with different air cavities to improve the bandwidth of sound absorption. Gai [13] combined L-shaped cavity with MPP to enhance the low frequency sound-absorptive performance. Tang et al. [14] found that small perforations on the face-sheet and honeycomb corrugation could improve the sound absorption at low frequencies. Huang et al. [15] investigated the effect of incompletely partitioned cavities on the sound absorption of MPP and found that if the insertion period and the length of separators were arranged appropriately, the low-frequency sound-absorption performance could be much improved. Pfretzschner et al. [16] presented a new strategy, which was named “MIU”, to increase the sound-absorption bandwidth. The MIU is composed of one thick plate with large perforations and another thin plate with a very high perforation ratio. This combination could produce sound absorption over two or three octave bands. 

Substantial works have focused on introducing additional Helmholtz resonators [17,18,19] or mechanical vibration [20,21] coupled with the MPP to improve its low-frequency sound-absorption performance. The combination of electromechanical system [22], shunted circuit loudspeaker [23], or aluminum-electrode PVDF piezoelectric film [24] with the MPP is also presented to improve the sound-absorption performance, especially in the low frequency range. 

Another alternative approach for improving the low-frequency sound-absorption performance of MPP in a limited space is increasing the depth of apertures, using attached tubes [25,26,27,28,29,30]. Among them, Li et al. proposed multiple perforated plates with extended tubes (PPETs) with the MPP or porous material to obtain broadband sound absorption in the low- to mid-frequency range in a constrained space. 

Some authors also investigated the sound-absorptive performance of perforated plates or micro-perforated plates periodically aligned in diffuse fields [31,32]. These studies showed that the sound-absorption performance under oblique incidence could differ from that under normal incidence. Moreover, the design guideline for diffuse sound absorption for low- to mid-frequency range is also a major concern in engineer noise control. Hence, the sound-absorption performance of PPETs with porous materials in a diffuse field requires further investigation, and in this paper, the diffuse sound absorption is theoretically and experimentally analyzed for absorbers composed of periodically arranged PPETs and porous materials.

## 2. Theoretical Model

A two-dimensional absorber composed of three parallel-arranged PPETs with porous material is considered. Figure 1 shows the basic module of a two-dimensional periodic absorber: G_1_, G_2_ and G_3_ are the surface admittances of three PPETs and G_4_ is the surface admittance of porous material. A plane wave, *p_i_* impinges on the absorber at the pitch angle of *θ* and azimuthal angle of *β*. The cavity wall and the rigid backing of the PPET (as shown in Figure 2) are considered to be acoustically rigid. In this study, the effect of the vibration of perforated panel is not considered for the sake of simplicity. The period of the composite absorber in both *x* and *y* directions is represented by *T*, as shown in Figure 1. The tube lengths, inner diameters and cavity depths of the three PPETs are 3.3, 10 and 100 mm, respectively, and the perforations of PPET1–3 are set as 0.90%, 1.54% and 2.59%, respectively.

### 2.1. Surface Impedance of the Perforated Plate with Extended Tubes

One unit of two-dimensional periodic PPET absorber are plotted in Figure 2. In the schematic diagram, *S_m_* denotes to the cross-sectional area of this unit; *t* and *d*_0_ respectively correspond to the length and inner diameter of the extended tubes; *r*_0_ and *r*_1_ are respectively the inner and outer radius of the extended tubes; and *t_p_* and *t*_0_ respectively correspond to the thickness of the cavity wall and the perforated plate. *S*_0_ = *πr*_0_^2^ corresponds to the inner cross-sectional area of the extended tubes, *S_a_* denotes the effective cross-sectional area of the back cavity, and *S*_1_ = *πr*_1_^2^ corresponds to the outer cross-sectional area of the extended tube. The specific acoustic impedance was derived in Reference [27] of a PPET which has a cavity with the thickness of *D*.
(1)$$Z_{p}=\frac{Z}{\sigma_{p}\rho c}=r_{p}+j\omega m_{p}-j/\left(\delta \tan\left(\omega (D-t+t_{0})/c\right)+\left(\delta -\sigma^{\prime }\right)\tan\left(\omega (t-t_{0})/c\right)\right)$$
(2)$$r_{p}=\frac{32\eta t}{\sigma_{p}\rho c{d_{0}}^{2}}\left({\left(1+\frac{k^{2}}{32}\right)}^{1/2}+\frac{\sqrt{2}kd_{0}}{64t}\right)$$
(3)$$\omega m_{p}=\frac{\omega t}{\sigma_{p}c}\left(1+{\left(9+\frac{k^{2}}{2}\right)}^{-1/2}+0.85\frac{d_{0}}{t}\right)$$
where *η* denotes the viscosity of the air; *ω* denotes to the angular velocity; and *ρ* and *c* are the density and the sound speed in the air, respectively. “$k=d_{0}\sqrt{\omega \rho /4\eta }$” corresponds to the perforation constant of the PPET; “$\sigma_{p}=NS_{0}/S_{m}$” denotes the perforation ratio of the PPET; “$\delta =S_{a}/S_{m}$” denotes the expansion ratio of cross-sectional area from the back cavity to the PPET; and “$\sigma^{\prime }=NS_{1}/S_{m}$” denotes the ratio of the outer cross-sectional area of the extended tubes over that of the PPET (N is the number of extended tubes).

### 2.2. Surface Impedance of Porous Material

In the present study, an equivalent fluid model known as JCAL model is used to predict the equivalent density and modulus of the porous materials, and then the equivalent density $\rho_{eq}(\omega )$ and the equivalent modulus $K_{eq}(\omega )$ of the porous fluid are given as follows:(4)$$\rho_{eq}(\omega )=\frac{\alpha_{\infty }\rho_{0}}{f}\left[1-j\frac{\sigma f}{\omega \rho_{0}\alpha_{\infty }}\sqrt{1+\frac{4{\alpha_{\infty }}^{2}\eta \rho_{0}\omega }{\sigma^{2}\Lambda^{2}f^{2}}}\right]$$
(5)$$K_{eq}(\omega )=\frac{\gamma P_{0}/f}{\gamma -\left(\gamma -1\right)\left[1-j\frac{f\kappa }{{k_{0}}^{\prime }C_{p}\rho_{0}\omega }\sqrt{1+j\frac{4{{k_{0}}^{\prime }}^{2}C_{p}\rho_{0}\omega }{\kappa {\Lambda^{\prime }}^{2}f^{2}}}\right]}$$
where $P_{0}$ is the ambient mean pressure; ϕ is the porosity, ω is the angular frequency; $\rho_{0}$ is the viscosity of the fluid; “$\nu =\eta /\rho_{0}=P_{r}v^{\prime }$” is the kinematic viscosity of the fluid; $P_{r}$ is the Prandtl number of the air; and γ is the specific heat ratio. 

The wave number, $k_{s}$, and characteristic impedance, $Z_{s}$, of the equivalent fluid medium are calculated as follows:(6)$$Z_{s}=\sqrt{\rho_{eq}(\omega )K_{eq}(\omega )}$$
(7)$$k_{s}=\omega \sqrt{\rho_{eq}(\omega )/K_{eq}(\omega )}$$

When the angle of the sound impinges on the porous materials is θ, the surface impedance of the porous material with the thickness, *L*, is given by the following:(8)$$Z=-j\frac{Z_{s}k_{s}}{\rho ck_{sz}}\cot\left(k_{sz}L\right)$$
where $k_{sz}=\sqrt{{k_{s}}^{2}-{k_{x}}^{2}-{k_{y}}^{2}}=\sqrt{{k_{s}}^{2}-k^{2}\sin^{2}\left(\theta \right)}$, ρc and k are the characteristic impedance and wave number of the air.

Then, an indirect characterization approach is used to obtain the transport parameters of melamine foam samples as described by Panneton and Olny [33,34], and a direct characterization of the viscous static permeability is carried out via measuring the resistivity [35]. The viscous-inertial frequency response function $\rho_{eq}\left(\omega \right)$ and the thermal frequency response function $K_{eq}\left(\omega \right)$ are measured with the three-microphone impedance tube method [36,37]. When ϕ, $k_{0}$, $\rho_{eq}\left(\omega \right)$ and $K_{eq}\left(\omega \right)$ are known, $\alpha_{\infty }$, ${k_{0}}^{\prime }$, Λ and $\Lambda^{\prime }$ can be experimentally estimated through an analytical inversion based on the models proposed by Johnson et al. [38] and Lafarge et al. [39]. Parameters of the porous foams are listed in Table 1.

### 2.3. Prediction of the Diffuse-Field Sound-Absorption Properties of Periodic Absorber

First, the total sound pressure, including the incident sound wave and the scattered sound waves, is given as follows:(9)$$p\left(x,0\right)=p_{i}\left(x,0\right)+p_{r}\left(x,0\right)$$

The incident sound wave is calculated as follows:(10)$$p_{i}\left(x,z\right)=p_{0}e^{j(-xk_{x}-yk_{y}+zk_{z})}$$
where $k_{x,m}$, $k_{y,n}$ and $\gamma_{mn}$ are the wavenumbers in the *x*, *y* and *z* directions, respectively. According to reference [32], the scattered sound wave is expressed in terms of modal expansion.
(11)$$p_{r}=\sum_{m=-\infty }^{+\infty }\sum_{n=-\infty }^{+\infty }A_{mn}e^{j\left(-xk_{x,m}-yk_{y,n}-z\gamma_{mn}\right)}$$
where $A_{mn}$ is the unknown amplitude of the (m,n)th mode of the scattered wave. Since the scattered field is periodic in the *x* and *y* directions, the wave numbers of the (m,n)th mode of the scattered wave are as follows:(12)$$\begin{array}{l} k_{x,m}=k\sin\theta \cos\beta +2m\pi /T\\ k_{y,n}=k\sin\theta \sin\beta +2n\pi /T\\ \gamma_{mn}=-jk\sqrt{{\left(\sin\theta \cos\beta +m\lambda /T\right)}^{2}+{\left(\sin\theta \sin\beta +n\lambda /T\right)}^{2}-1} \end{array}$$
where λ=2π/k is the wavelength. The corresponding (m,n)th acoustic mode which could propagate to the far field must satisfy the following relationship: (13)$${\left(\sin\theta \cos\beta +m\lambda /T\right)}^{2}+{\left(\sin\theta \sin\beta +n\lambda /T\right)}^{2}\leq 1$$

Then, according to the relation between the sound pressure and particle velocity on the surface, we get the following:(14)ρcv(x,y,0)=−G(x,y)p(x,y,0)
where
(15)$$G\left(x,y\right)=\left\{ \begin{array}{l} G_{1}=\frac{1}{Z_{p1}},\;0\leq x<\frac{T}{2},\;0\leq y<\frac{T}{2}\\ G_{2}=\frac{1}{Z_{p2}},\;\frac{T}{2}\leq x<T,\;0\leq y<\frac{T}{2}\\ G_{3}=\frac{1}{Z_{p3}},\;0\leq x<\frac{T}{2},\;\frac{T}{2}\leq y<T\\ G_{4}=\frac{1}{Z_{p\mathrm{sam}}},\;\frac{T}{2}\leq x<T,\;\frac{T}{2}\leq y<T \end{array}\right.$$

Inserting Equations (9) and (15) into the boundary condition Equation (14), multiplying both sides by a factor of $e^{jm\left(2\pi /T\right)x}e^{jn\left(2\pi /T\right)y}$ and then integrating the equation with respect to *x* and *y* over the period *T* yields a set of linear algebraic equations:(16)$$\begin{array}{l} \sum_{m=-\infty }^{+\infty }\sum_{n=-\infty }^{+\infty }A_{mn}\left(g_{\left(m-m^{\prime },n-n^{\prime }\right)}+\delta_{\left(m-m^{\prime },n-n^{\prime }\right)}\left(\frac{\gamma_{mn}}{k_{0}}\right)\right)=P_{0}\left(\delta_{m,0}\delta_{n,0}\cos\theta -g_{\left(m,n\right)}\right)\\ m=-\infty ,\cdots ,+\infty \;\&\;n=-\infty ,\cdots ,+\infty \end{array}$$
where $g_{\left(m,n\right)}=\frac{1}{L_{x}L_{y}}\int_{0}^{L_{y}}\int_{0}^{L_{x}}G\left(x,y\right)e^{jm\left(2\pi /T\right)x}e^{jn\left(2\pi /T\right)y}dxdy$ and the Kronecker symbol, $\delta_{m,n}$, is defined as follows:(17)$$\delta_{\left(m-m^{\prime },n-n^{\prime }\right)}=\left\{ \begin{array}{l} 1,\;m-m^{\prime }=n-n^{\prime }\\ 0,\;m-m^{\prime }\neq n-n^{\prime } \end{array}\right.$$

This infinitely large system of equations will be terminated at the index limits “*m,n* = ±*2*N*”, where *N* is the number of elements in one period, and in this study, *N = 2*. By solving the above equations, the coefficients $A_{mn}$ can be obtained. According to Mechel [40], the angle-dependent sound-absorption coefficient α(θ,β) is given by Equation (18):(18)$$\alpha \left(\theta ,\beta \right)=1-{\left|\frac{A_{00}}{P_{0}}\right|}^{2}-\frac{1}{\cos\theta }\times \sum_{m\neq 0}{\sum_{n\neq 0}\left|\frac{A_{mn}}{P_{0}}\right|}^{2}\sqrt{1-{\left(\sin\theta \cos\beta +m\frac{\lambda }{T}\right)}^{2}-{\left(\sin\theta \sin\beta +n\frac{\lambda }{T}\right)}^{2}}$$
where the summation runs over all the radiating harmonics only, the second term is the specular reflection and the third term is the scattered sound. The averaged diffuse-field sound-absorption coefficient is defined as follows [32,41]:(19)$$\alpha_{s}=\frac{1}{2\pi }\int_{0}^{2\pi }\left(\int_{0}^{\pi }\alpha_{\left(\theta ,\beta \right)}\sin\left(2\theta \right)d\theta \right)d\beta$$

When the period width *T* << λ, the specular reflection is only non-evanescent reflection. In this case, the sound-absorption coefficient and normalized impedance of the periodic absorber can be derived as follows:(20)$$\alpha \left(\theta ,\beta \right)=1-{\left|\frac{A_{00}}{P_{0}}\right|}^{2}$$
(21)$$Z\left(\theta ,\beta \right)=\frac{1-\frac{A_{00}}{P_{0}}}{1+\frac{A_{00}}{P_{0}}}$$

## 3. Oblique-Incidence Sound-Absorption Properties of Periodic Absorber

Figure 3a shows the sound-absorption coefficients of a periodic absorber composed of units of three parallel-arranged PPETs and porous material at different incident angles (*θ*), when *T* = 10 cm, *β* = 0°, and the frequency interval in the calculation is 10 Hz. In the following discussion, the porous material chosen in the simulation is the Basotect TG foam (BASF(China) Co. Ltd., Beijing, China). The period length, *T*, is much smaller than the wavelength of the incident wave, so the sound-absorption coefficient and the normalized impedance can be calculated by using Equations (20) and (21), respectively. Three absorption peaks observed at 160, 210 and 275 Hz are owning to the Helmholtz resonances of the PPETs, while the fourth peak, observed at 800 Hz, is the resonance frequency provided by the porous material.

The influences of the incident angle, *θ*, on the specific resistance and reactance are shown in Figure 3b,c, respectively. It is observed that the incidence angle can greatly influence the sound-absorption coefficients of four parallel-arranged PPETs. The sound-absorption coefficient gets larger when the incidence angle is increased from 0° to 60°; it then decreases when the incidence angle goes from 60° to 89°, and it drops greatly at a near-grazing incidence angle (*θ* = 89°). Figure 3b shows that, at a larger incidence angle, the characteristic resistance and the reactance (absolute value) of the proposed absorber are smaller. It is clear that, when the incidence angle, *θ*, is equal to 0°, the specific resistance is much larger than 1, and the sound-absorption coefficient is reduced. When the incidence angle is *θ* = 60°, the normalized acoustic impedance matches well with the characteristic impedance of the air, and the sound-absorption coefficient is largely increased. As shown in Figure 3b, when incidence angle is *θ* = 89°, the normalized acoustic resistance is too small compared with the characteristic impedance of the air, so the sound-absorption coefficient is smaller. Hence, for larger sound absorption at oblique incidence, the normalized resistance of the proposed absorber should be larger than 1.

Figure 4 shows the variation of period length on the sound-absorption coefficient of the periodic absorber. When the period length, *T* (10 cm), is much smaller than the wavelength of the incident wave, the parallel sound absorption of the periodic absorber could be maintained. When the period length of the absorber is increased to 40 cm, the absorption coefficient drops abruptly at 840 Hz, corresponding to which the wavelength is comparable to the period length, *T*. When the period length of the absorber is increased to 80 cm (which is comparable to the wavelength corresponding to 420 Hz), the absorption coefficient drops abruptly at around 420 Hz. It could be concluded that the period of the proposed periodic absorber is a critical factor that controls the mechanism of parallel absorption. Hence, in a diffuse sound field, the period length of the periodic absorber should not be larger than the wavelength of interest.

The effect of the sound-absorption coefficient of the periodic absorber with the azimuthal angle *β* is illustrated in Figure 5. It is observed that the azimuthal angle has much less influence on the diffuse sound absorption compared with that of the incidence angle. When the period length, *T*, is 20 cm, the sound-absorption coefficient varies a little with the azimuthal angle in the high frequency range and remains the same in the low frequency range. When the period length, *T*, is 40 cm, the variation of the sound-absorption coefficient with azimuthal angle is more significant in the high frequency range.

## 4. Experimental Validation

The diffuse-field sound-absorption coefficient of parallel-arranged PPETs and porous material (BASF(China) Co. Ltd., Beijing, China) is measured in a reverberation room, and the measured results are compared with the theoretical predictions. The measurements are conducted in compliance with the ISO 354-2003 standard [42]. The three dimensions of the cubic chamber are 6.86, 4.94 and 5.79 m, respectively, and the total area of the test specimen is 10.3 m^2^. Several curved sheets as reflectors are hung in the room. The excitation speaker is a spherical sound source. Figure 6 shows the photo picture of test sample, and the period length of the absorber, *T*, is 80 cm, the cavity depth of the parallel-arranged PPETs is 7 cm and the thickness of the porous material is 10 cm. The perforated panels used in the experiments were made of plastic plate, and the extended tubes were made of copper. Two kinds of Basotect foam, Basotect G+ and Basotect TG, were chosen for the experiments, and the parameters are listed in Table 1.

The measured and predicted 1/3 octave sound-absorption coefficient curves of three parallel-arranged PPETs and porous foam are shown in Figure 7 and Figure 8. The sound-absorption peaks of PPETs are weakened due to 1/3 octave resolution. A reasonable agreement could be found in most frequency band, except below 200 Hz. The discrepancy mainly comes from the too-low modal density of the reverberation chamber in low frequency. The PPETs combined with porous material in a limited thickness of 10 cm have a good sound-absorption performance between 160 and 3150 Hz.

## 5. Conclusions

The low-frequency sound-absorptive properties in diffuse field are predicted and measured for the periodically arranged sound absorbers composed of perforated plates with extended tubes (PPETs) and porous materials. The results show that increasing the incidence angle will decrease the characteristic resistance of the periodic absorber, and the sound-absorption coefficient reaches the lowest near the grazing incidence angle. The effect of the azimuthal angle on the sound-absorption coefficient is found to be insignificant. It is noted that the period of the proposed absorber is a critical factor that controls the mechanism of parallel absorption, and when the period of absorber arrangement is comparable to the wavelength of incident wave, the sound-absorption coefficient drops abruptly. Both theoretical and measured results show that periodically arranged PPETs combined with porous material could keep good sound-absorptive performance in low- to mid- frequency in diffuse sound field.

## Figures and Tables

**Figure 1 materials-13-01091-f001:**
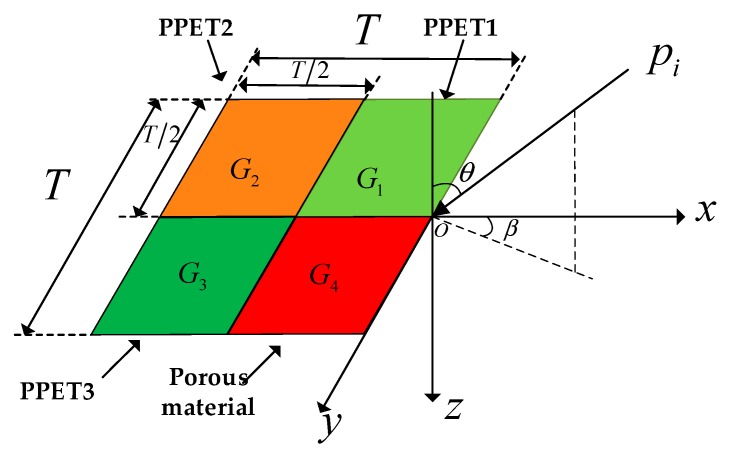
Schematic diagram of parallel-arranged perforated plates with extended tubes (PPETs). The angles of the incidence and azimuthal sound are defined as *θ* and *β* in this schematic.

**Figure 2 materials-13-01091-f002:**
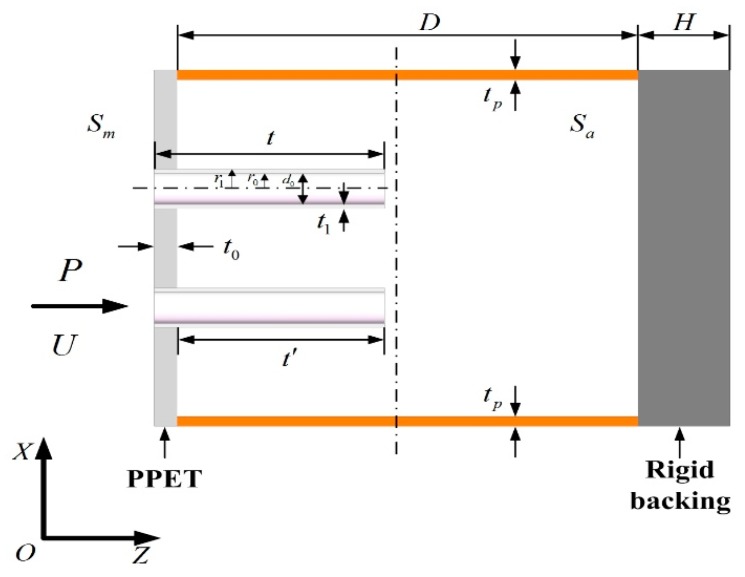
One unit of two-dimensional periodic absorber composed of three parallel-arranged PPETs with porous material.

**Figure 3 materials-13-01091-f003:**
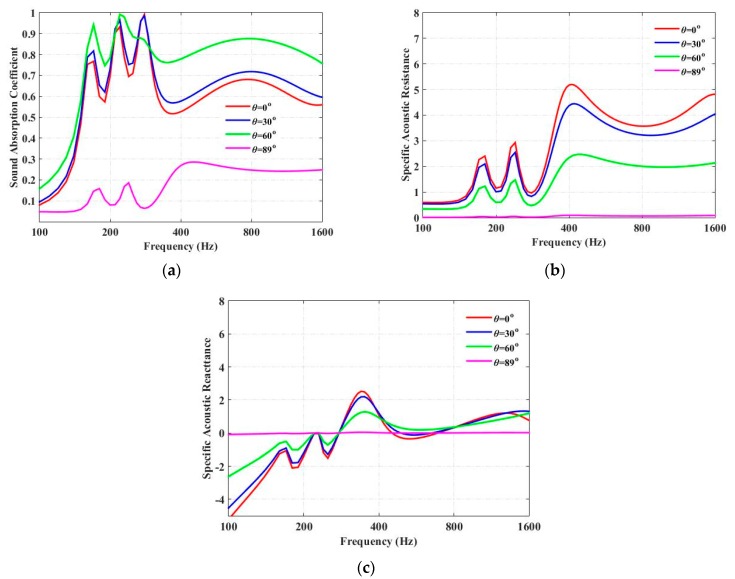
Comparison of the sound-absorption coefficients of the unit composed of three parallel-arranged PPETs and porous material at different angles when *T* = 10 cm, *β* = 0°. (**a**) Sound-absorption coefficient; (**b**) specific acoustic resistance; (**c**) specific acoustic reactance.

**Figure 4 materials-13-01091-f004:**
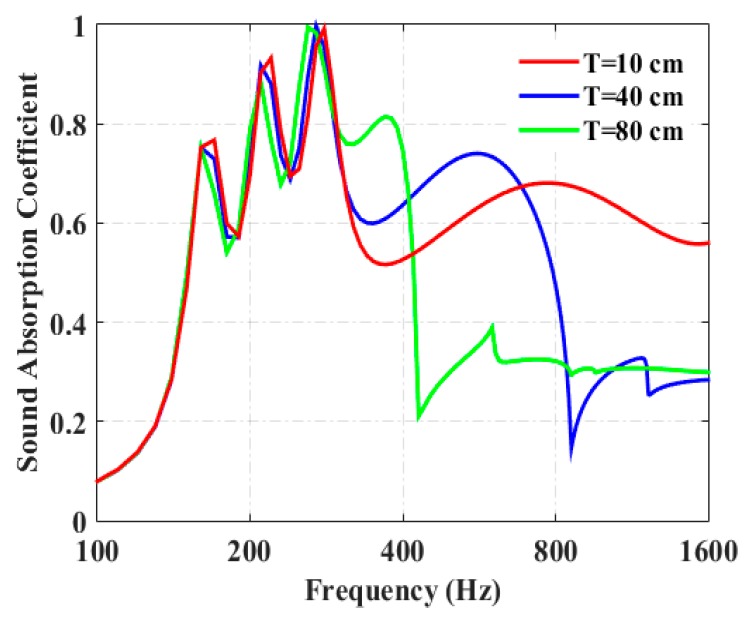
Influence of the period length on the sound-absorption coefficient of periodic absorber, when *θ* = *β* = 0°.

**Figure 5 materials-13-01091-f005:**
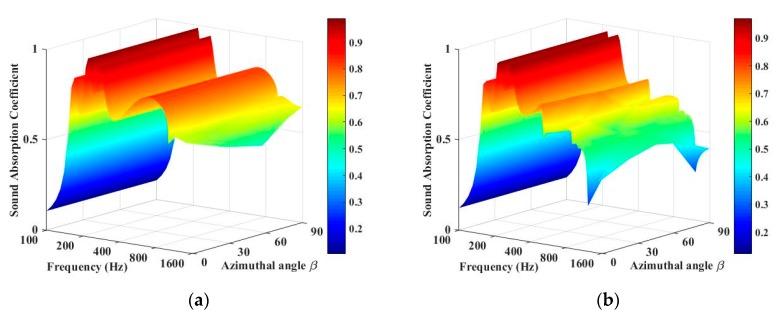
Sound-absorption coefficients of periodic absorber at different azimuthal angles. (**a**) *T* = 20 cm, *θ* = 45°. (**b**) *T* = 40 cm, *θ* = 45°.

**Figure 6 materials-13-01091-f006:**
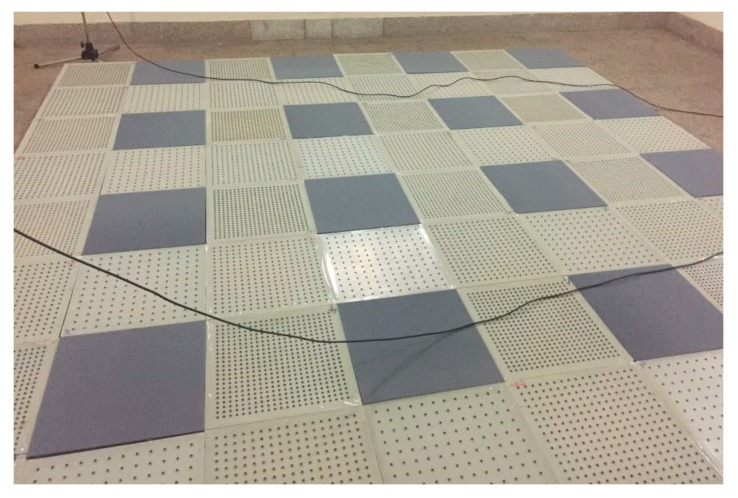
Picture of test sample.

**Figure 7 materials-13-01091-f007:**
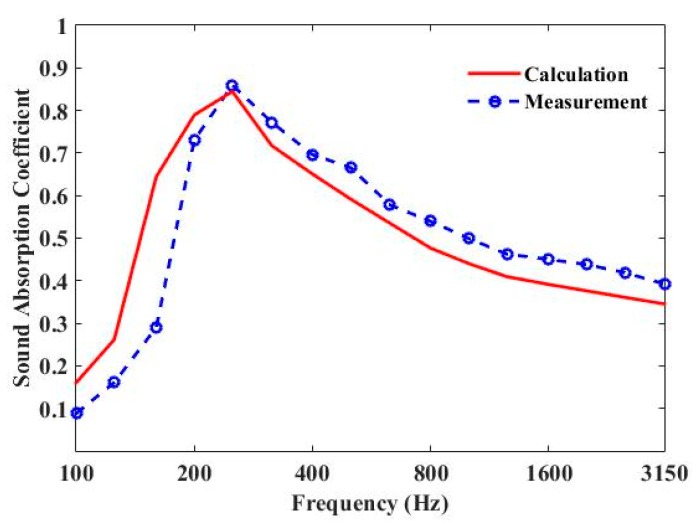
Comparison of the measured and calculated sound-absorption coefficient for periodic absorber composed of units of three parallel-arranged PPETs and Basotect G+ foam.

**Figure 8 materials-13-01091-f008:**
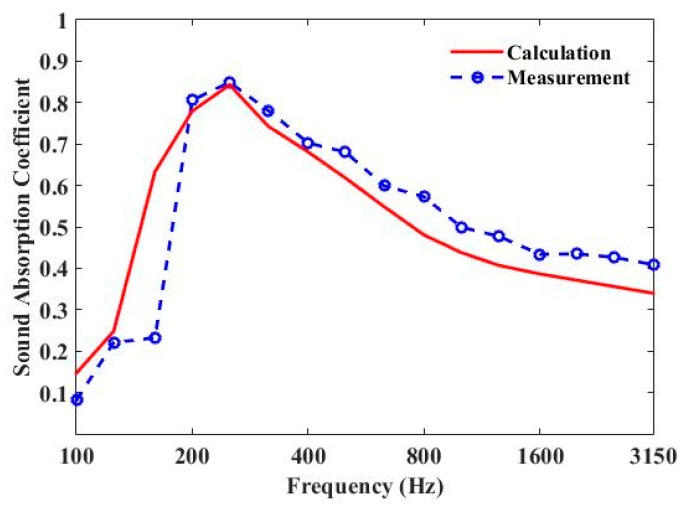
Comparison of the measured and calculated sound-absorption coefficient of periodic absorber composed of units of three parallel-arranged PPETs and Basotect TG foam.

**Table 1 materials-13-01091-t001:** Parameters for the Basotect G+ and TG foam.

Parameters	*σ* (N·m^−4^·s)	ϕ	α_∞	Λ (μm)	Λ′ (μm)	${k^{\prime }}_{0}\left(\times 10^{-10}\;m^{2}\right)$
Basotect G+	10934 ± 182	0.994	1.04 ± 0.03	92 ± 5	197 ± 9	27 ± 1
Basotect TG	7800 ± 200	0.993	1.03 ± 0.02	134 ± 16	317 ± 32	47 ± 4

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
