# Peer review of "Diffuse Sound Absorptive Properties of Parallel-Arranged Perforated Plates with Extended Tubes and Porous Materials"

_materials, 2020, doi:10.3390/ma13051091_

Round 1

Reviewer 1 Report

The present paper deals with the diffuse field sound absorption coefficient of periodically-arranged absorbers composed of inhomogeneous perforated plates with extended tubes and porous materials in both experimentally and theoretically. The reviewer thinks that the present theoretical model has a serious problem on the assumption for treating the extended reaction behavior of the absorber especially for large value of T compared to the wavelength of incident sound wave. This should be solved for publication of the present manuscript. Also, the following points should be considered.

  1. Line53, it should define clearly that what the words “M”, “P”, “M” stand for in the word “M-P-M”.
  2. Section 2.2 Regarding the porous material model, it should be mentioned explicitly that the present study uses an equivalent fluid model with JCA model as a porous fluid model.
  3. Section 2.2, Regarding Equation (8), this is for surface impedance at normal incidence. Why did the authors use the normal incidence value? The reason should be included within manuscript clearly. Also, symbols Z_s and k_s should be defined.
  4. In Figure 3, please describe in the caption what the three figures respectively stand for.
  5. Figure 4, the results of T=40 cm and 80 cm. In this cases, locally reacting assumption used in the present model does not hold and the results is doubtful. The results should calculate with extended reacting model. This is also true for comparison with measured values in Section 4 where T = 80 cm.
  6. Section 4, the ISO 354-2003 standard requires to show the results from 100 Hz to 5000 Hz. Why the authors present the results below 3150 Hz? The reason must be clearly stated. Also, the information of reverberation chamber is necessary especially about the room shape, used sound source and diffuser treatment.

Author Response

Authors’ Response to the Reviewers’ comments

Dear editor,

Thank you and the reviewers for your valuable suggestions. We have carefully considered each comment made by the reviewers and made significant revisions in the paper.

Our responses to the reviewer’s questions and the detailed changes are listed below. We greatly appreciate your time and efforts to improve our manuscript for publication and we look forward to hearing from you soon.

Sincerely,

    Dengke Li and Daoqing Chang

Comments from the Reviewer 1: 

The present paper deals with the diffuse field sound absorption coefficient of periodically-arranged absorbers composed of inhomogeneous perforated plates with extended tubes and porous materials in both experimentally and theoretically. The reviewer thinks that the present theoretical model has a serious problem on the assumption for treating the extended reaction behavior of the absorber especially for large value of T compared to the wavelength of incident sound wave. This should be solved for publication of the present manuscript. Also, the following points should be considered.

1) Page 1, Line53, it should define clearly that what the words “M”, “P”, “M” stand for in the word “M-P-M”.

Reply: Many thanks for the comments, and we made some corrections in line 53-55 that “A triple layer absorber composed of permeable membrane (PM) [10] and the double-leaf MPP (DLMPP) was proposed to improve the absorption performance, and the peak absorption at resonance was found to become significantly higher.”

2) In Section 2.2, Regarding the porous material model, it should be mentioned explicitly that the present study uses an equivalent fluid model with JCA model as a porous fluid model.

Reply: Many thanks for the comments, it is now added in line 127-128 that “In the present study, an equivalent fluid model known as JCAL model is used to predict the equivalent density and modulus of the porous materials.”

3) In Section 2.2, Regarding Equation (8), this is for surface impedance at normal incidence. Why did the authors use the normal incidence value? The reason should be included within manuscript clearly. Also, symbols Z_s and k_s should be defined.

Reply: Many thanks for the comments. In the revised manuscript we already corrected above errors. We have rechecked the formules and calculations, the surface impedance at oblique incidence is

 (1)

where  , and  are the characteristic impedance and wave number of the air, respectively.  and  are respectively the wave number and characteristic impedance of the porous materials.

Figure 1 illustrates thesound absorption coefficients of the periodic absorber at different incident angles when,base on the different surface impedance equation of porous material(Solid line-based on the old impedance model of porous material; Dased line-based one the revised impedance model of porous material). Figure 2 shows the comparison of sound absorption characteristics in the diffuse sound field based on the two models. The difference between the results obtained based on the two models is very tiny.

(a)

(b)

(c)

Figure 1. Comparison of the sound absorption coefficients of the unit composed of 3parallel-arranged PPETs and porous material at different angles when,. (a) Sound absorption coefficient; (b) Specific acoustic resistance; (c) Specific acoustic reactance. Solid line-based on the old impedance model of porous material; Dased line-based one the revised impedance model of porous material.

Figure 2. Comparison of the calculated averaged diffuse-field sound absorption coefficient for periodic absorber composed of units of 3 parallel-arranged PPETs and porous material, blue solid line represents the old impedance model, and red dotted line represents the revised impedance model .

4) In Figure 3, please describe in the caption what the three figures respectively stand for.

Reply: Many thanks for the comments. In the revised manuscript we already corrected above errors.

5) Figure 4, the results of T=40 cm and 80 cm. In this cases, locally reacting assumption used in the present model does not hold and the results is doubtful. The results should calculate with extended reacting model. This is also true for comparison with measured values in Section 4 where T = 80 cm.

Reply: Many thanks for the comments. In the theoretical modeling, we treat the extended tubes as a local reacting resonator, and we use the local impedance of the PPETs as the input impedance of the periodic PPETs. When calculating the admitance G of each periodic PPET, the internal structure of the unit, such as the size of the perforations, must meet the low frequency limit. In this study, the three PPETs working at low frequencies, and the resonance frequency of the four PPETs is f=165Hz, 210Hz, 275Hz, respectively. Meanwhile, the hole spacing () between the perforations of the three PPETs is respectively 0.03m, 0.02m and 0.018m, which is much less than the wavelength of interest. The period length T mentioned in the article refers to the spatial distribution of the macroscopic sound absorption structure. For large value of T compared to the wavelength, the calculation model of the acoustic characteristics of the periodic structure under random incidence conditions is also applicable, see reference [1-3].

The autors also used the present model to predict the sound absorption of the period absorbers with T=160cm. The comparison between the calcualtion and measured results are illustrated in Figure 3. It is observed that the present model also compare well with the test result. Increasing the period length T will decrease the peak values of the absorption, however the main sound absorption frequency band change slightly.

(a)

Figure 3. Comparison of the measured and calculated averaged diffuse-field sound absorption coefficient for periodic absorber composed of units of 3 parallel-arranged PPETs and Basotect G+ foam. (a) Blue solid line and blue circle dotted line respectively represent the calculation and test result with period length T=160 cm.

6) Section 4, the ISO 354-2003 standard requires to show the results from 100 Hz to 5000 Hz. Why the authors present the results below 3150 Hz? The reason must be clearly stated. Also, the information of reverberation chamber is necessary especially about the room shape, used sound source and diffuser treatment.

Reply: Many thanks for the comments. In the revised manuscript we already corrected above errors.  

In the present paper, we are focused on the low- to mid-frequency sound absorption of the periodic PPET-Porous material absorber. It is also shown in Figure 2 that the frequency bands of sound absorption coefficient above 0.4 from 160Hz to 1600 Hz. So we measured the diffuse sound absorption in the frequency range from 100 Hz to 3150 Hz.

In the revised manuscript we added that “ The three dimensions  of the cubic chamber is 6.86 m, 4.94 m, 5.79 m, respectively. And the total area of the test specimen is 10.3. Several curved sheets as reflectors are hung in the room. The excitation speaker is a spherical sound source. ”

REFERENCES

[1] H. Drotleff, R. Wack, P. Leistner. Absorption of Periodically Aligned Absorber Strips in Concrete Structures. Building Acoustics, 16, 233-236 (2009).

[2] C. Wang, L. Huang,Y. Zhang, Oblique incidence sound absorption of parallel arrangement of multiple micro-perforated panel absorbers in a periodic pattern, Journal of Sound and Vibration 333,  6828-6842 (2014).

[3] Mechel, F.P. Sound Fields at Periodic Absorbers. Journal of Sound and Vibration 136, 379-412 (1990).

The authors would like to thank the editor and the reviewers very much for the comments. These comments are very helpful to improve the quality of this manuscript.

Reviewer 2 Report

The paper presents the results of theoretical and experimental studies of sound absorption in perforated plates with extended tubes and porous materials. Though the study is of good quality in general, the authors do not show comparison with other kinds of absorbers and do not give the reader with the numerics to demonstrate that the proposed configuration provides improved characteristics compared to others. The latter is very important for the readers and it should be mainly addressed in the revision including careful checking of the text of the manuscript.
Some other minor comments are below.
In Figure 1, it is not evident where angle beta is determined.
Table 1 is more or less abundant since the only column has dissimilar values.
Figure 5 looks quite careless.

Author Response

Authors’ Response to the Reviewers’ comments

Dear editor,

Thank you and the reviewers for your valuable suggestions. We have carefully considered each comment made by the reviewers and made significant revisions in the paper.

Our responses to the reviewer’s questions and the detailed changes are listed below. We greatly appreciate your time and efforts to improve our manuscript for publication and we look forward to hearing from you soon.

Sincerely,

    Dengke Li and Daoqing Chang

Comments from the Reviewer 2: 

The paper presents the results of theoretical and experimental studies of sound absorption in perforated plates with extended tubes and porous materials. Though the study is of good quality in general, the authors do not show comparison with other kinds of absorbers and do not give the reader with the numerics to demonstrate that the proposed configuration provides improved characteristics compared to others. The latter is very important for the readers and it should be mainly addressed in the revision including careful checking of the text of the manuscript.

Reply: Many thanks for the comments. The authors strongly agree to the reviewer that the superiority of proposed configuration to others is of great importance. This paper is mianly focused on the diffuse sound absorption properties of the periodic PPET absorber. A numerical comparison of the averaged diffuse-field sound absorption between proposed absorber with a typical MPP absorber is done in follows. It could be found from Figure 1 that the proposed absorber have much improved low frequency sound absorption (below 400 Hz) compared with that of a single MPP absorber. And the proposed absorber could achieve a wide band frequency range from 100-3150Hz in a limited thickness of 100mm.

Figure 1. Comparison of the calculated averaged diffuse-field sound absorption coefficient between periodic absorber with MPP absorber. The parameters of MPP are as follows: , , D=10 cm, and .

Some other minor comments are below.

In Figure 1, it is not evident where angle beta is determined.

Reply: Many thanks for the comments. It is now added in the revised manuscript (line 102) that “The angles of the incidence and azimuthal sound are defined as   and   in this schematic.”

Table 1 is more or less abundant since the only column has dissimilar values.

Reply: Many thanks for the comments. It is added in the revised manuscript (line 99) that “The tube lengths, inner diameters and cavity depths of the three PPETs are 3.3mm, 10mm and 100mm, respectively. And the perforations of PPET1-3 are set as 0.90 %, 1.54 % and 2.59 %, respectively.”

Figure 5 looks quite careless.

Reply: Many thanks for the comments. In the revised manuscript we already corrected above errors. 

The authors would like to thank the editor and the reviewers very much for the comments. These comments are very helpful to improve the quality of this manuscript.

Round 2

Reviewer 1 Report

The corrections made in the revised manuscript are acceptable. I recommend now for publishing this article.

Reviewer 2 Report

In general, the authors corrected their manuscript in accordance with reviewers' comments.